DOI: 10.1038/s41467-018-02834-8　　**OPEN**

# The tumour microenvironment creates a niche for the self-renewal of tumour-promoting macrophages in colon adenoma

Irene Soncin[1], Jianpeng Sheng[1], Qi Chen[1], Shihui Foo[2], Kaibo Duan[2], Josephine Lum[2], Michael Poidinger [2], Francesca Zolezzi[2,3], Klaus Karjalainen[1] & Christiane Ruedl[1]

Circulating CCR2[+] monocytes are crucial for maintaining the adult tissue-resident F4/80[hi]MHCII[hi] macrophage pool in the intestinal lamina propria. Here we show that a subpopulation of CCR2-independent F4/80[hi]MHCII[low] macrophages, which are the most abundant F4/80[hi] cells in neonates, gradually decline in number in adulthood; these macrophages likely represent the fetal contribution to F4/80[hi] cells. In colon adenomas of Apc[Min/+] mice, F4/80[hi]MHCII[low] macrophages are not only preserved, but become the dominant subpopulation among tumour-resident macrophages during tumour progression. Furthermore, these pro-tumoural F4/80[hi]MHCII[low] and F4/80[hi]MHCII[hi] macrophages can self-renew in the tumour and maintain their numbers mostly independent from bone marrow contribution. Analyses of colon adenomas indicate that CSF1 may be a key facilitator of macrophage self-renewal. In summary, the tumour microenvironment creates an isolated niche for tissue-resident macrophages that favours macrophage survival and self-renewal.

[1] School of Biological Sciences, Nanyang Technological University, 60 Nanyang Drive, Singapore 637551, Singapore. [2] Singapore Immunology Network, Agency for Science, Technology and Research (A*STAR), 8A Biomedical Grove, Singapore 138648, Singapore. [3] Present address: GALDERMA R&D, 06902 Sophia Antipolis Cedex France. Klaus Karjalainen and Christiane Ruedl jointly supervised this work. Correspondence and requests for materials should be addressed to K.K. (email: Klaus@ntu.edu.sg) or to C.R. (email: Ruedl@ntu.edu.sg)

Macrophages comprise a heterogeneous population of tissue-resident immune cells that contribute to tissue homeostasis, support the host defence system and can impact on the initiation and propagation of several diseases, including cancer (reviewed in refs.[1,2]). The majority of tissue-resident macrophages are established prenatally[3–5] and are mainly independent from any further haematopoietic input due to their capacity to self-renew in situ to maintain population size[6–10]. For example, microglia, the resident macrophage population of the central nervous system, develop early during embryonic development from yolk sac precursors and effectively self-maintain in situ throughout adulthood[5,6,11]. Although the majority of the tissue-resident macrophages are maintained independently from bone marrow (BM)-derived monocytes, numerous studies have demonstrated that macrophages of certain tissues, including the dermis[5,12], mammary gland[13], heart[14], pancreas[15] and intestine[5,16,17], require the input of monocytes to retain their pool during adulthood. In particular, the intestinal macrophages depend on the constant and fast replenishment of circulating blood Ly6C$^{hi}$ monocytes, not only during inflammation but also under normal healthy conditions[16–18]. In a healthy unperturbed colon, monocytes attracted to the intestine gradually differentiate into tissue-resident macrophages by losing Ly6C expression, up-regulating the expression of macrophage markers, such as CX3CR1, F4/80, CD64 and CD11c[16,19], and secreting or responding to the anti-inflammatory cytokine interleukin-10 [20,21]. Mice lacking either the chemokine receptor CCR2 or its ligand CCL2 have reduced numbers of intestinal macrophages[17], thus implying that the homeostatic recruitment of macrophage precursors to the gut lamina propria (LP) may be dependent on a CCR2–CCL2 axis. Upon intestinal inflammation, tissue-resident macrophages are still derived from circulating monocytes, but convert from being anti-inflammatory macrophages to highly Toll-like receptor-responsive inflammatory cells[16].

Tissue-resident macrophages are also naturally present in various tumours, including lung and mammary tumours where macrophages are monocyte-derived[22,23] or even self-renewing[24]. The developmental origins and the maintenance kinetics of resident macrophages in intestinal tumours, however, have not been studied thus far.

Here we show that the colon LP of normal adult mice contains a major population of F4/80$^{hi}$MHCII$^{hi}$ macrophages and a population of gradually disappearing F4/80$^{hi}$MHCII$^{low}$ cells. Both F4/80$^{hi}$ macrophage populations are relatively slowly replenished by BM-derived cells, particularly MHCII$^{low}$ cells, compared to F4/80$^{int}$ monocyte/macrophage cells that have a fast turnover. The tumour microenvironment, however, is enriched with F4/80$^{hi}$MHCII$^{low}$ macrophages. Furthermore, while F4/80$^{hi}$MHCII$^{low}$ macrophages do not require CCR2$^+$ monocytes for their maintenance in the healthy gut LP or in tumours, we found that F4/80$^{hi}$MHCII$^{hi}$ macrophages become independent from CCR2$^+$ monocytes in tumours only, potentially due to tumour-derived colony stimulating factor-1 (CSF1) that supports macrophage self-renewal. Moreover, we demonstrate that CCR2-independent intratumoural F4/80$^{hi}$ macrophages, and not the CCR2-dependent monocytes or monocyte-derived macrophages, support cancer progression, suggesting this particular F4/80$^{hi}$ cell fraction as an attractive therapeutic target.

## Results

### Skewed representation of myeloid cells in colon adenomas. We first profiled colon LP myeloid subpopulations from wild-type (WT) C57BL/6J mice, previously treated with dextran sodium sulphate (DSS), and compared them with their tumoural counterparts obtained from the polyps collected from DSS-treated $Apc^{Min/+}$ mice (Fig. 1). Similar as previously reported[5], colon LP CD45$^+$ cells comprised three major myeloid cell fractions (I–III) after excluding CD11c$^{hi}$MHCII$^{hi}$ dendritic cells: F4/80$^{hi}$CD11b$^+$ tissue-resident (tumour-resident in polyps) macrophages (fraction I), F4/80$^{int}$CD11b$^+$ (fraction II) and F4/80$^{low}$CD11b$^+$ neutrophils (fraction III) (Fig. 1a). The majority (>85%) of fraction I cells expressed high levels of major histocompatibility complex (MHC) class II and only a small proportion (~5%) of cells were low for MHC class II expression. Fraction II could be further subdivided into four distinct subpopulations in a waterfall-shaped distribution on a Ly6C vs. MHCII dot plot consisting of three distinct monocyte-differentiation stages (P1: Ly6C$^{hi}$MHCII$^-$; P2: Ly6C$^{hi}$MHCII$^+$; P3: Ly6C$^-$MHCII$^+$) and one eosinophil fraction (Ly6C$^-$MHCII$^-$) (Fig. 1a). We then analysed the presence of these subpopulations in DSS-accelerated colon tumours from $Apc^{Min/+}$ mice. The same LP populations were also identified in colon polyps but with markedly different frequencies (Fig. 1b). In these tumours, neutrophils (fraction III) were the most abundant myeloid cell type (>40%) followed by monocyte-derived cells (fraction II) and tumour-resident macrophages (fraction I). Interestingly, eosinophils in tumours represented only a minor fraction (~3%) of the total CD11b$^+$ cells. Furthermore, tumour-resident macrophages (fraction I) were clearly enriched in MHCII$^{low}$ cells (Fig. 1a,b), which were almost neglectable in normal adult colon LP.

Characterization of myeloid subpopulations in the colon LP of untreated C57BL/6J mice and in "spontaneously" formed tumours obtained from 5-month-old $Apc^{Min/+}$ mice confirmed the presence of all myeloid subpopulations described in Fig. 1, although with a clear increase in the proportion of the tumoural CD11b$^+$F4/80$^-$ neutrophil fraction (up to 70% of total CD11b$^+$ cells) (Supplementary Fig. 1).

### MHCII$^{low}$ macrophages accumulate during tumour progression. A previous study reported that F4/80$^{hi}$ tissue-resident macrophages in the colon LP progressively disappear with increasing age[17]. We therefore analysed the myeloid cell subsets present in the fetal colon (embryonic day 19.5) and compared them to those present in the colon LP of young and old mice (from 1 week to 12 months). In contrast to Bain et al.,[17] our analysis clearly found that all three macrophage fractions (I–III) were maintained across all age groups with only minor fluctuations detected in their frequency (Fig. 2a). The predominant tissue-resident macrophage subset in the fetal colon was the F4/80$^{hi}$MHCII$^{low}$ fraction. However, as a function of age this macrophage subset almost disappeared to leave mainly F4/80$^{hi}$MHCII$^{hi}$ cells to represent the colon tissue-resident macrophages (fraction I) (Fig. 2b). This decrease in MHCII$^{low}$ cell representation suggests that the LP environment gradually changes with age, perhaps due to the establishment of a mature microbiota and/or a tonic increase in inflammatory mediators[17].

Given the high frequency of F4/80$^{hi}$MHCII$^{low}$ cells in colon polyps (Fig. 1b), we assessed whether there was an association between tumour progression and the frequency of MHCII$^{low}$ macrophages. Therefore, for this analysis we opted for the spontaneous tumour model of $Apc^{Min/+}$ mice, which generate progressively adenomas up to 5–6 mm in size that is rarely achieved in the DSS-accelerated tumour model. Indeed, we observed an increasing abundance of this cell fraction in conjunction with tumour progression until MHCII$^{low}$ macrophages became a dominant cell type among tumour-resident macrophages (Fig. 2c). This observation clearly suggests that the tumour microenvironment favours the differentiation or maintenance of MHCII$^{low}$ macrophages.

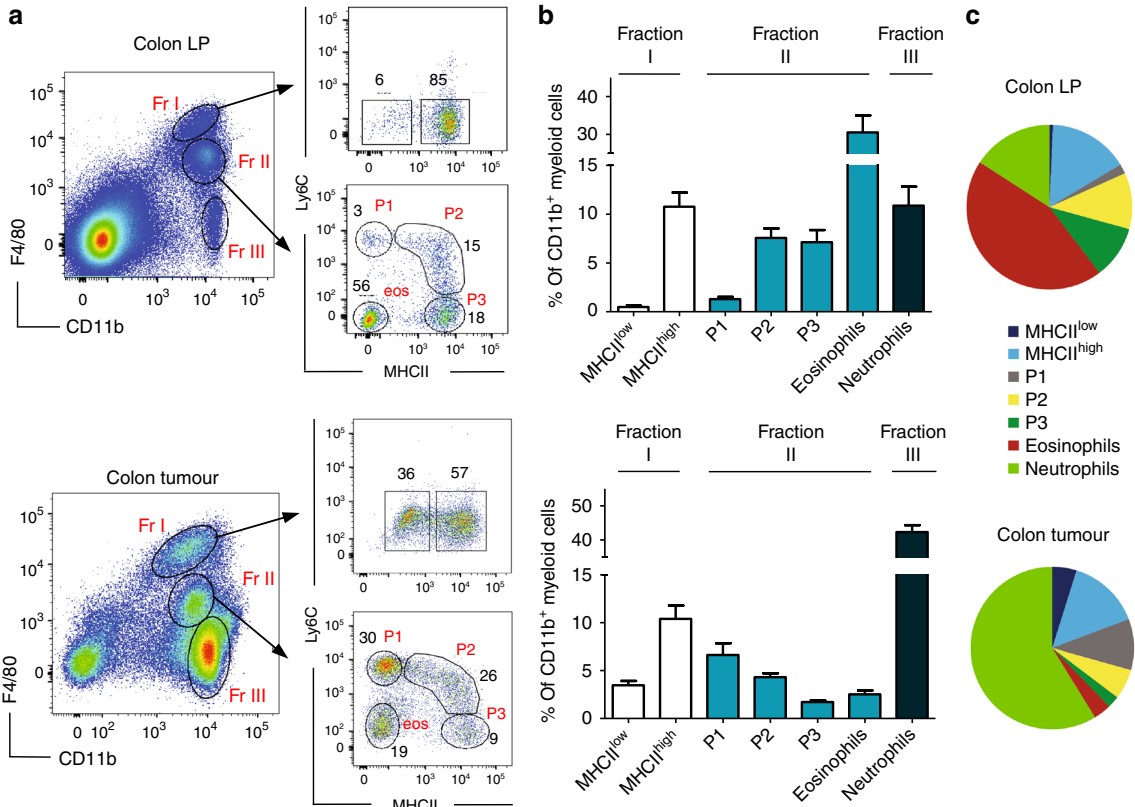

**Fig. 1** Myeloid cell heterogeneity in colon lamina propria and colon tumours. WT C57BL/6J and $Apc^{Min/+}$ mice aged 6 weeks old were administered drinking water supplemented with dextran sodium sulphate for 7 days, and analysed 4 weeks later. **a** Flow cytometry representative dot plots of colon lamina propria (LP) (upper panel) and tumour cell subpopulations (lower panel). Three different myeloid fractions (I–III) are defined by the differential expression of F4/80. Fraction I represents F4/80$^{hi}$ tissue-resident macrophages, which can be further subdivided into MHCII$^{hi}$ and MHCII$^{low}$. Fraction II contains monocytes (P1), two monocyte-derived macrophage subpopulations (P2 and P3) and eosinophils (eos), based on differential expression of MHCII and Ly6C. Fr III consists of neutrophils. Gating strategy is shown in Supplementary Fig. 4. **b** Bar charts of the distinct myeloid cell subpopulations obtained from the colon LP ($n = 8$) (upper bar chart) and 2–3 mm colon tumours ($n = 20$) (lower bar chart). White bars: fraction I; light blue bars: fraction II and black bars: fraction III. Error bars represent the s.e.m. **c** Pie charts show the proportions of F4/80$^{hi}$ tissue-resident macrophages (MHCII$^{hi}$ and MHCII$^{low}$), monocytes (P1), monocyte-derived macrophages (P2–P3), neutrophils and eosinophils across colon LP and tumours

**Intratumoural macrophages are independent from CCR2$^+$ monocytes**. Given that circulating Ly6C$^{hi}$ monocytes express high levels of CCR2 receptor[25], we decided to examine the presence of this receptor on tissue-resident and monocyte-derived macrophages in colon LP and adenoma polyps. The aim of this experiment was to determine the potential origins of these cells. In the LP, consistent with being descendants of classical monocytes, Ly6C$^{hi}$MHCII$^-$ (P1), Ly6C$^{hi}$MHCII$^{hi}$ (P2) and Ly6C$^-$MHCII$^+$ (P3) cells in fraction II subset had the highest and most homogeneous expression of CCR2 (Fig. 3a)[17], whereas tissue-resident F4/80$^{hi}$MHCII$^{hi}$ and F4/80$^{hi}$MHCII$^{low}$ macrophages were highly heterogeneous in their CCR2 expression levels, from very high to very low levels (Fig. 3a and Supplementary Fig. 2). As expected, numbers of monocytes and monocyte-derived cells (P1, P2 and P3) in fraction II were severely diminished in $Ccr2^{-/-}$ mice (Fig. 3b). In addition, MHCII$^{hi}$ tissue-resident macrophages were also mostly CCR2 dependent, whereas MHCII$^{low}$ cells were not, which is likely a reflection of their markedly lower CCR2 expression level (Fig. 3b).

Interestingly, we found that tumour-resident MHCII$^{hi}$ and MHCII$^{low}$ macrophages both expressed relatively low levels of CCR2 compared to their LP counterparts and, consistently, their numbers did not change in $Ccr2^{-/-}$ mice. Conversely, fraction II cells (P1, P2 and P3) in polyps behaved exactly like their WT counterparts (Fig. 3a, c).

Taken together, these data assert that the tumour microenvironment can modify the properties of MHCII$^{hi}$ tissue-resident macrophages such that they also become CCR2 independent like MHCII$^{low}$ cells in the LP and hence are presumably able to maintain themselves without monocyte replenishment.

**Turnover of colon lamina propria myeloid cells**. We utilized the $Kit^{MerCreMer}$/R26 fate mapping mouse[5]—where yellow florescence protein (YFP) expression can be induced in early BM progenitors—to monitor the cell population turnover rates driven by the BM input in colon LP myeloid cell populations. After injecting adult mice with tamoxifen, the labelling index of tissue-resident macrophages (F4/80$^{hi}$MHCII$^{low}$ and MHCII$^{hi}$), monocytes (P1), monocyte-derived macrophages (P2–P3 fractions), eosinophils and neutrophils was analysed at different time points over a period of 5 months. As anticipated, all tested myeloid populations, with the exception of tissue-resident macrophages, were rapidly labelled with YFP and reached a plateau by 1–2 weeks after the final tamoxifen injection. These data suggest rapid replacement of myeloid cell populations by BM-derived cells (Fig. 4).

Although it is widely considered that gut F4/80$^{hi}$MHCII$^{hi}$ cells lose their prenatal origin and are replaced by CCR2$^+$ monocytes in adulthood[16], our data clearly demonstrated that they are in fact

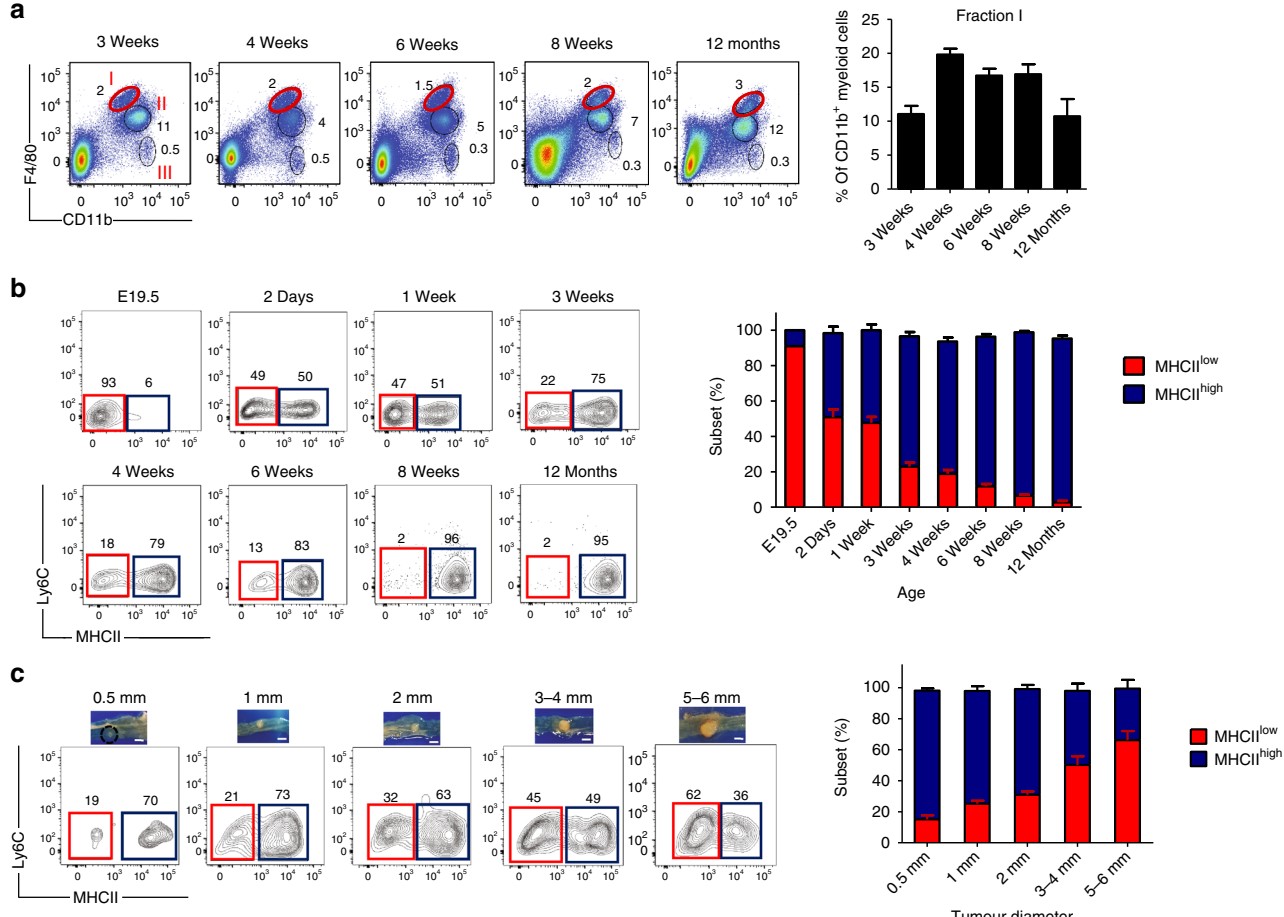

**Fig. 2** Ageing and tumour progression influence the ratio between F4/80[hi]MHCII[hi] and F4/80[hi]MHCII[low] subpopulations. **a** F4/80[hi] tissue-resident macrophages persist in the colon LP after birth into adulthood, as shown by the representative flow cytometry dot plots and bar chart. Error bars represent the s.e.m. **b** The frequency of colon F4/80[hi]MHCII[low] cells rapidly declines after birth. Representative dot plots of F4/80[hi]MHCII[hi] and F4/80[hi]MHCII[low] subpopulations obtained from fetal colon (E19.5) and from colons of mice aged 2 days, 1, 3, 4, 6 and 8 weeks and 12 months. The bar chart represents the age-dependent ratio between F4/80[hi]MHCII[hi] (blue) and F4/80[hi]MHCII[low] (red) subpopulations in the colon LP. E19.5: $n = 1$ (pool of 15 embryos obtained from 3 different pregnant mice); day 2: $n = 2$ (each group pool of 5 mice); 1 week: $n = 2$ (each group pool of 5 mice); 3 weeks: $n = 6$ mice; 4 weeks: $n = 5$ mice; 6 weeks: $n = 4$; 8 weeks: $n = 7$ and 12 months: $n = 4$ mice. Error bars represent the s.e.m. **c** Representative flow cytometry analysis (left panel) and mean percentage of F4/80[hi]MHCII[hi] and F4/80[hi]MHCII[low] subpopulations (right panel) in tumours of different sizes (0.5–6.0 mm in diameter) obtained from $Apc^{Min/+}$ mice. 0.5 mm: $n = 3$; 1 mm: $n = 3$; 2 mm: $n = 4$; 3–4 mm: $n = 3$; 5–6 mm: $n = 4$. Error bars represent the s.e.m. Scale bars: 2.5 mm. Gating strategy is shown in Supplementary Fig. 4

replaced by BM-derived monocytes only very slowly, if at all. Our fate mapping data indicated that it took several months to fully replace F4/80[hi]MHCII[hi] cells, whereas MHCII[low] macrophages exhibited minimal labelling (Fig. 4), suggesting that these tissue-resident macrophages retained their original fetal seed population for a long period.

**Intratumoural resident macrophages expand by self-renewal.** We noted that F4/80[hi]MHCII[low] tumour-resident macrophages in particular seemed to expand in polyps and thus wanted to determine if this effect was due to increased self-renewal or the recruitment of new cells. Again, our adult fate mapping analyses in $Kit^{MerCreMer/R26}Apc^{Min/+}$ mice showed that the recruitment of new cells to DSS-accelerated adenomas was not increased and that the population turnover rate of F4/80[hi]MHCII[low] cells was comparable to that of the LP (Fig. 5a). A decrease in YFP labelling was observed in tumour-resident F4/80[hi]MHCII[hi] cells when compared to their LP counterparts. This is probably due to the fact that the labelling index of macrophages in polyps was

"frozen" to the level corresponding to the time for their entrapment in this niche that became independent from new inputs (Fig. 4 and Fig. 5a).

Of note, when tamoxifen was administered during embryogenesis (Fig. 5b, upper panel), all myeloid cell subpopulations—including tissue-resident macrophages in the healthy LP and colon tumours—exhibited similar YFP tagging patterns (Fig. 5b, lower panel). This result was expected, as all of these cells are the progeny of classical haematopoietic stem cells, as previously described[5].

We found clear evidence, however, for increased self-renewal of F4/80[hi] macrophages that could explain their capacity to expand in polyps. Firstly, tumour-resident macrophages were clearly in an active phase of the cell cycle unlike those in the LP, as indicated by the expression of the Ki-67 cellproliferation marker (Fig. 6a). Secondly, RNA-sequencing (RNA-seq) analyses of cell-sorted, purified tumour-resident F4/80[hi] (both MHCII[hi] and MHCII[low]) cells showed a clear upregulation of several cell cycle regulators, including cyclin-dependent kinase Cdk1, cyclin-A2 and -B2, E2f2, NEK family of serine/threonine

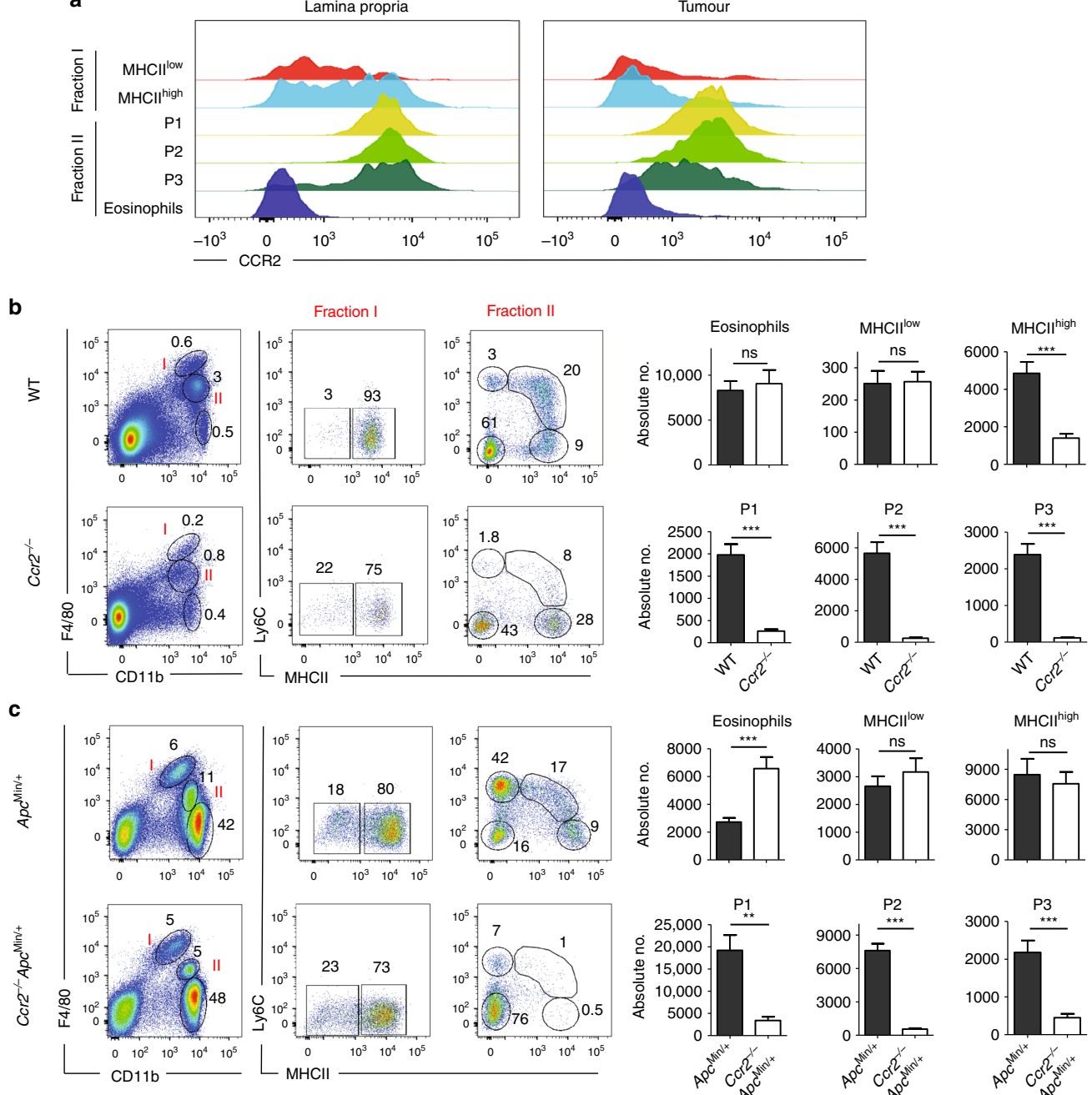

**Fig. 3** Lamina propria and intratumoural colon F4/80hiCD11b+ macrophage subsets show differential CCR2-driven monocyte dependence. **a** Fluorescence-activated cell sorting histograms showing CCR2 expression profiles on distinct colon LP and intratumoural myeloid cell subpopulations: fraction I consists of MHCIIlow and MHCIIhi cells, and fraction II consists of monocytes (P1) and monocyte-derived macrophages (P2 and P3). **b** Myeloid cell profiling in WT and $Ccr2^{-/-}$ colon LP. Representative flow cytometry analysis with F4/80 and CD11b-expressing myeloid subpopulations, obtained from the colon LP of WT and $Ccr2^{-/-}$ mice. Fractions I and II were further dissected for MHCII and Ly6C expression (left panel). The absolute numbers of eosinophils, F4/80hiMHCIIhi and F4/80hiMHCIIlow tissue-resident macrophages and P1–P3 subpopulations are shown (right panel). The bar chart represents the mean number of mice in each group and the error bars represent the s.e.m. (WT; $n = 7$ and $Ccr2^{-/-}$; $n = 14$). **c** Myeloid cell profiling in WT and $Ccr2^{-/-}$ colon tumours as described above for LP cells. Bar charts show the mean±s.e.m. of absolute numbers of eosinophils, F4/80hiMHCIIhi and F4/80hiMHCIIlow tissue-resident macrophages and P1–P3 subpopulations obtained from $Apc^{Min/+}$ ($n = 8$) and $Apc^{Min/+}Ccr2^{-/-}$ ($n = 9$) mice. Statistical significance was determined using an unpaired Student's $t$-test. **P<0.001; ***P < 0.0001; ns, not significant. Gating strategy is shown in Supplementary Fig. 4

kinases and various cell division cycle and mini-chromosome maintenance proteins family members, in comparison to their LP counterparts (Fig. 6b). Interestingly, we could not find any evidence for active cell cycling of F4/80hiMHCIIhi and F4/80hiMHCIIlow cells in the LP (Fig. 6a, b). This effect could be because F4/80hiMHCIIlow macrophages gradually disappear in the adult LP and probably do not vigorously self-renew, and that F4/80hiMHCIIhi cells are predominantly maintained in the LP by recruited BM-derived cells and, therefore, do not depend on in situ self-renewal.

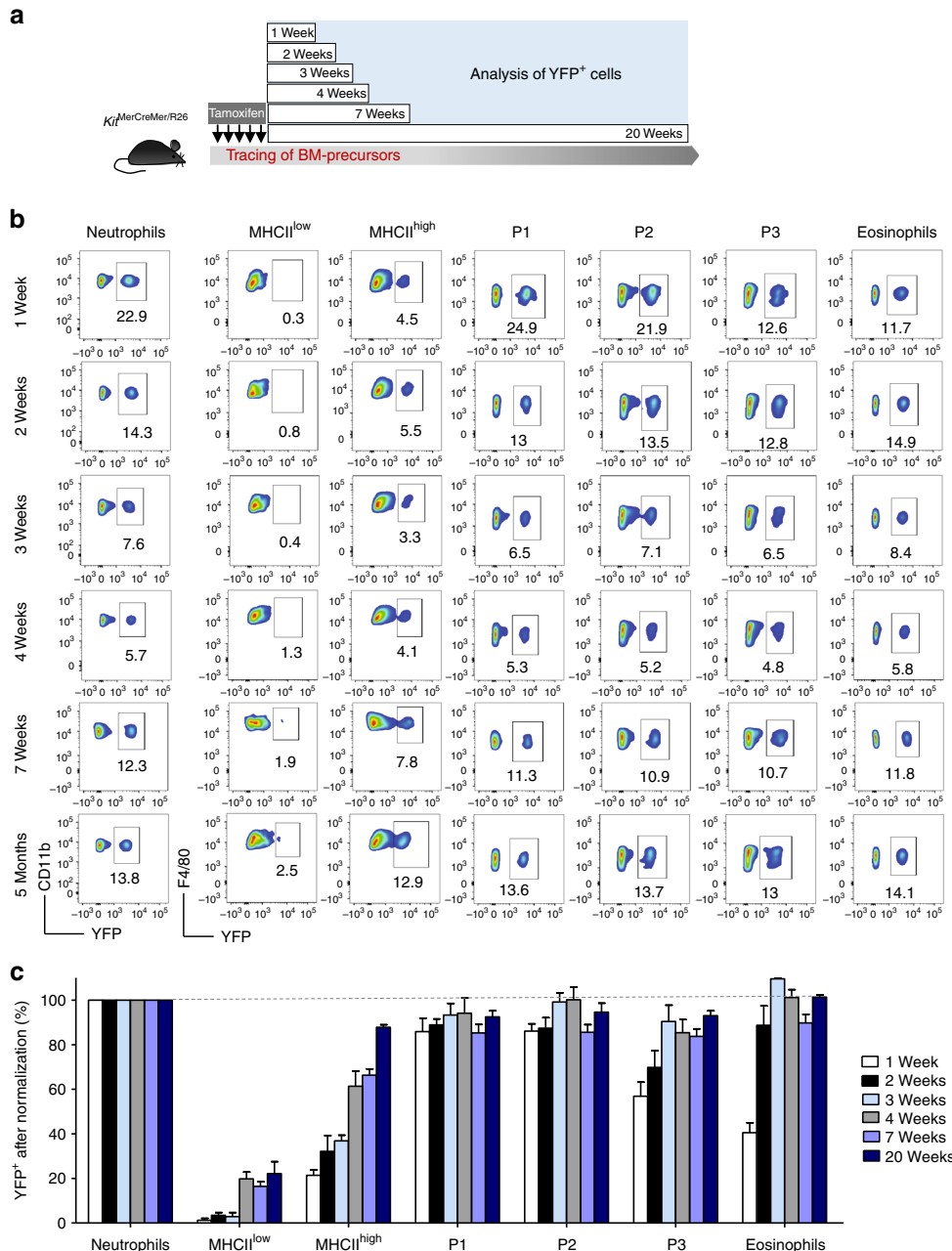

**Fig. 4** Intestinal tissue-resident macrophages exhibit a slower, gradual replacement by bone marrow-derived cells compared to other gut myeloid cells. **a** Schematic representation of the adult fate mapping protocol using $Kit^{MerCreMer/R26}$ mice. Mice aged 6 weeks were injected with tamoxifen five times and groups of 4–8 animals were sacrificed 1, 2, 3, 4, 7 and 20 weeks later. **b** Representative flow cytometry analysis indicating the labelling efficiency of distinct colon myeloid cell populations as defined in Fig. 1: MHCII^low-expressing and MHCII^hi-expressing tissue-resident macrophages, monocytes and monocytes-derived macrophages (P1–P3), eosinophils and neutrophils. Neutrophils acted as internal controls for labelling efficiency and the tracings are from the same mouse. Gating strategy is shown in Supplementary Fig. 4. **c** The bar chart represents the mean percentage of yellow florescence protein-positive (YFP+) cells after normalization to the percentage of YFP+ neutrophils. The error bars represent the s.e.m.

**Tumour microenvironment alters resident macrophage phenotype.** Besides the upregulation of genes involved in cell proliferation, macrophages in polyps clearly adopted a distinct metabolic signature. Similar to cancer cells, which can alter their metabolism in aerobic glycolysis and production of lactate (so-called Warburg effect) macrophages in polyps increased the transcript levels of key glycolytic genes, such as ENO1, GAPDH, TPI1, PGAM and LDHA, with the F4/80^hiMHCII^low macrophages expressing the highest levels (Fig. 6c). In addition, our RNA-seq found that both tissue-resident subsets displayed enhanced transcripts of ARG1, typical for M2-polarized and tumour-associated macrophages (TAMs)[26,27], with MHCII^low macrophages being the major producers of ARG1 within tumours (Fig. 6c–e). The expression of other urea cell cycle genes, such as Arginase-2 (ARG2) and Arginosuccinate synthetase 1 (AAS1), was increased in tumour-resident macrophages as well (Fig. 6c). Moreover, several transcripts of matrix metalloproteases (MMPs), such as MMP2, MMP9 and MMP12, involved in the degradation of extracellular matrix and promotion of metastasis were upregulated in both subpopulations of tissue-resident macrophages when compared to their LP counterparts (Fig. 6f), suggesting a possible contribution of these cells in tumour progression.

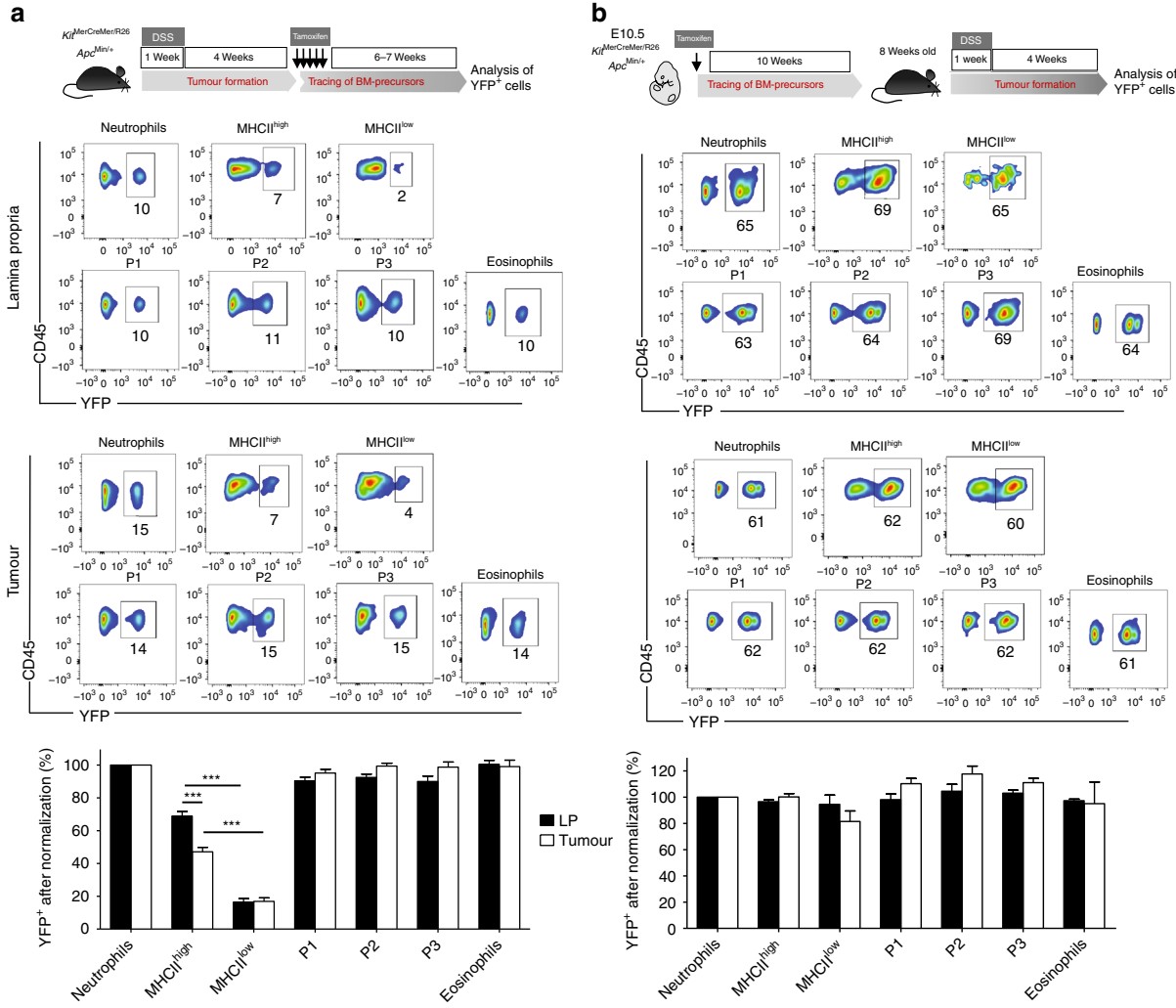

**Fig. 5** Identification of self-renewing colon macrophage subsets in healthy and tumour tissue. **a** Dextran sodium sulphate (DSS)-induced intestinal adenoma formation was combined with adult fate mapping using the $Kit^{MerCreMer/R26}$ and $Kit^{MerCreMer/R26}Apc^{Min/+}$ mice (schematic representation shown in upper panel). Representative flow cytometry analysis illustrates the colon myeloid cell subset labelling efficiency. Gating strategy is shown in Supplementary Fig. 4. The bar chart represents the mean percentage of yellow fluorescence protein-positive (YFP+) cells after normalization to the percentage of YFP+ neutrophils±s.e.m. A total of 15 $Kit^{MerCreMer/R26}Apc^{Min/+}$ and 20 $Kit^{MerCreMer/R26}$ 6–8-week-old mice were subjected to DSS treatment. Statistical significance was determined by two-way ANOVA followed by Bonferroni test; ***$P < 0.001$. For reasons of clarity, the non-significant differences between groups were not indicated. **b** Embryonic fate mapping confirms the prenatal origins of intestinal LP and tumoural tissue-resident F4/80hiMHCIIhi and F4/80hiMHCIIlow macrophages. Pregnant $Kit^{MerCreMer/R26}$ and $Kit^{MerCreMer/R26}Apc^{Min/+}$ mice (E10.5) were administered tamoxifen by intraperitoneal injection, as schematically illustrated in the upper panel. The offspring were then subjected to DSS treatment at 8 weeks old and sacrificed 4 weeks later for tissue and cell isolation. Representative flow cytometry analysis shows the labelling efficiency of each myeloid cell subset analyzed. Gating strategy is shown in Supplementary Fig. 4. The bar chart represents the mean percentage of YFP+ cells after normalization to the percentage of YFP+ neutrophils. The error bars represent the s.e.m. A total of 4 $Kit^{MerCreMer/R26}Apc^{Min/+}$ and 6 $Kit^{MerCreMer/R26}$ mice were analysed. E, embryonic day; P1, monocytes; P2 and P3, monocyte-derived macrophages. Statistical significance was determined by two-way ANOVA followed by Bonferroni test. For reasons of clarity, the non-significant differences between groups were not indicated

Of note, although hierarchical clustering between distinct colon myeloid cells clearly identified F4/80hiMHCIIhi and F4/80hiMHCIIlow as demarcated populations, "high/low" pairs always formed more closely related couples in tumours or in the LP than "low/low" or "high/high" comparisons between samples. Although these data suggested a close relationship between F4/80hiMHCIIhi and F4/80hiMHCIIlow cells (Fig. 6g), ingenuity pathway analysis (IPA®) revealed that multiple pathways involved in glycolysis, gluconeogenesis, urea cycle, hypoxia-inducible factor-1 and colorectal cancer metastasis signalling were significantly upregulated in F4/80hiMHCIIlow cells ($P < 0.05$ by right-tailed Fisher's exact test, Supplementary Fig. 3, red arrows). Clearly, tumour-resident F4/80hiMHCIIlow cells not only accumulate, but also

undergo a robust metabolic reprogramming during tumour progression.

**CSF1 maintains and expands pro-tumoural resident macrophages.** CSF1 is a crucial growth factor for macrophage proliferation and survival, and supports the self-maintenance of tissue-resident macrophages[26–29]. Here, we compared the levels of CSF1 in the colon LP and colon adenomas and detected a progressive increase in the amounts of CSF1 as a function of polyp size (Fig. 7a). In addition, when the effects of CSF1 were neutralized upon exposure to an anti-CSF1 receptor neutralizing antibody in tumour-bearing $Apc^{Min/+}$ mice, we observed a strong

reduction in the number of both F4/80[hi]MHCII[low] and F4/80[hi]MHCII[hi] tumour-resident macrophages, whereas the numbers of monocytes and monocyte-derived macrophages were unperturbed (Fig. 7b). All together, these data strongly suggest that F4/80[hi]-resident macrophages exposed to high levels of CSF1 (and likely other tumour niche factors) in the tumour micro-environment gain the ability to self-renew within the tumour.

To evaluate the pro-tumoural potential of different types of tumour-resident macrophages, polyp numbers and their sizes were assessed in $Ccr2^{-/-}Apc^{Min/+}$ and anti-CSF1R injected $Apc^{Min/+}$ mice. Clearly, CCR2 deficiency, although strongly reduced monocytes and monocyte-derived macrophage P2 and P3 populations, had no effect on colonic polyp numbers in $Ccr2^{-/-}Apc^{Min/+}$ mice compared to their $Apc^{Min/+}$ littermates in the DSS-accelerated tumour model (Fig. 7c, upper panel). In contrast, depletion of tumour-resident F4/80[hi] macrophages lowered significantly the polyp counts in the colon ($P < 0.05$ by unpaired

Student's $t$-test) and the tumours formed were of smaller size ($P < 0.001$ by unpaired Student's $t$-test) suggesting their pro-tumoural role in established tumours (Fig. 7c, lower panel).

## Discussion

The macrophages found resident in the majority of tissues are established during fetal development and are able to locally self-maintain in adulthood without any further input from BM haematopoietic progenitors[4]. In tissues exposed to a low-grade inflammation, however, such as the intestine and skin, or to mechanical stress, such as the heart, the maintenance of these cells relies not on self-renewal, but on a continuous replenishment by infiltrating BM-derived CCR2-dependent Ly6C[hi] circulating monocytes[12,14,17].

Here we confirmed that in adult mice the vast majority of colon LP macrophages are derived from CCR2-dependent BM progenies. These macrophages included F4/80[hi]MHCII[hi] LP-

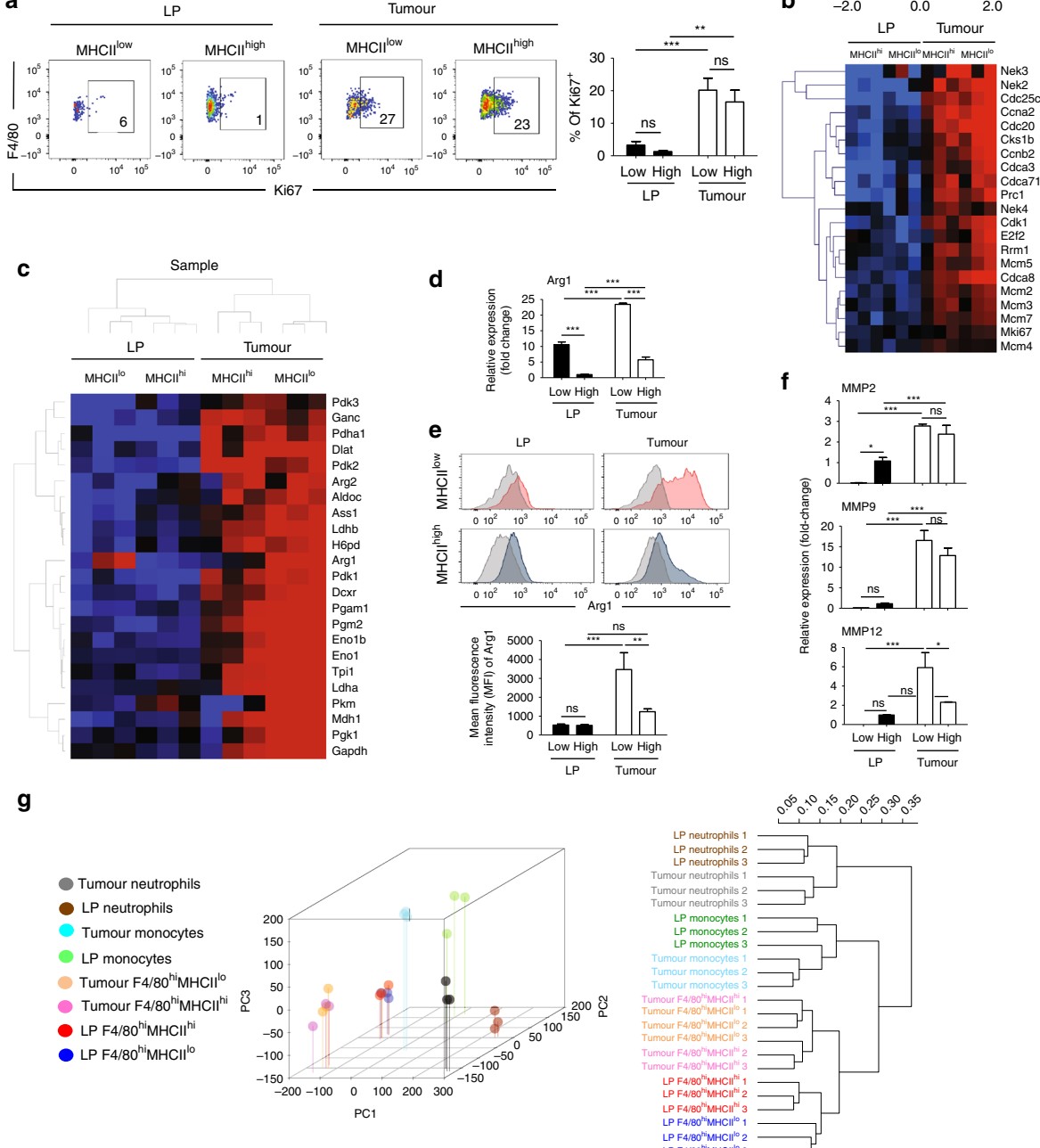

resident macrophages (fraction I) in addition to a classical monocyte-derived series of F4/80$^{int}$CD11b$^+$ cells (fraction II). We found one exception to this rule, whereby LP-resident F4/80$^{hi}$MHCII$^{low}$ macrophages that could be detected in high numbers only in young mice were also clearly found in adult Ccr2$^{-/-}$ mice. Interestingly, our fate mapping experiments showed that LP-resident F4/80$^{hi}$ macrophages exhibited a very slow turnover rate compared to other myeloid populations in the LP. In particular, F4/80$^{hi}$MHCII$^{low}$ macrophages showed minimal labelling after adult mouse tamoxifen administration, which suggests that they are a fetal-derived population that gradually disappear with age. This phenomenon is perhaps expected due to increased competition to occupy the niche by monocyte-derived cells, which can then efficiently differentiate into F4/80$^{hi}$MHCII$^{hi}$ macrophages[17].

Similarly to the gut, an analogous tissue-resident macrophage subpopulation that expresses low levels of MHCII was recently characterized in the skin dermis—another tissue exposed to commensal microbes and under tonic, low-level inflammation[30]. Consistent with the colon tissue-resident macrophages, the dermal F4/80$^{hi}$MHCII$^{hi}$ fraction is clearly BM derived, whereas the recruitment of the F4/80$^{hi}$MHCII$^{low}$ subset in the dermis occurs mainly during embryogenesis and rapidly declines during adulthood[5,12].

To date, the developmental origins of intestinal TAMs and their maintenance in colon adenomas have not been extensively studied. As macrophages have become potential targets for cancer therapy[31], we therefore decided to study the dynamics of myeloid populations in colon adenomas. In breast cancer[13], glioma[32] and lung metastases[33] TAMs can arise from Ly6C$^{hi}$ circulating monocytes. Their influx and positioning in the tumour is mediated by chemokines (e.g., CCL2), growth factors (e.g., CSF1) and complement components[31,34]. In addition, a recent study demonstrated that local proliferation of TAMs contributes to their accumulation in spontaneous mammary tumours[24]. Many have speculated that in cancers exposed to microbes, such as colon cancer, the recruitment of macrophages and their maintenance might differ from cancers located in "sterile" sites, such as breast cancers[31].

Here, we have analysed in detail the origins and maintenance of macrophage subpopulations in colon tumours obtained from Apc$^{Min/+}$ mice, which is a widely used animal model for intestinal

tumourigenesis[35], and focused on both subsets of tissue-resident F4/80$^{hi}$ macrophages in the colon LP.

The lymphoid and myeloid composition of colon adenomas of Apc$^{Min/+}$ mice differed from that of the colon LP. In the LP, we found that lymphoid cells (mainly B cells) were the predominant cell type, whereas in adenomas the majority consisted of myeloid cells and were almost devoid of lymphocytes and, notably, eosinophils that were the most common LP myeloid cells. Neutrophils and monocytes were strongly increased in adenomas, but interestingly also F4/80$^{hi}$MHCII$^{low}$ macrophages. These MHCII$^{low}$ cells are usually a minor population in LP, but clearly "flourished" in the tumour environment in our model to such an extent that in large polyps they became the dominant population among F4/80$^{hi}$ cells. Our fate mapping studies and analyses of Ccr2$^{-/-}$ mice showed that MHCII$^{low}$ macrophages expanded in situ in adenomas by self-renewal, without any recruitment of new cells. This finding was also supported by our RNA-seq and Ki-67 analyses showing that MHCII$^{low}$ macrophages in polyps were actively cycling, unlike macrophages found in the colon LP. Similarly, the tumour micro-environment was also conducive for the self-renewal of F4/80$^{hi}$MHCII$^{hi}$ macrophages. This subpopulation of macrophages was also actively cycling and had become independent from CCR2-dependent inputs from the BM. The tumour milieu, however, did not favour monocyte differentiation to macrophages. In adenomas, monocytes (P1) got "stuck" at the beginning of the "waterfall" maturation pathway to macrophages (P3), which resulted in the F4/80$^{hi}$ cells becoming the dominant macrophage population. This effect has also been reported in the context of mammary cancers[13]. Our data clearly suggest that unique characteristics of the tumour niche can trigger self-renewal of F4/80$^{hi}$ macrophages. CSF1 is certainly a key component of the tumour niche to manipulate macrophage self-renewal, but other cues can also contribute to this process and, therefore, reconstruction of the tumour niche would be very illuminating.

According to the "M1/M2 paradigm", mainly based on in vitro polarization experiments, pro-inflammatory M1macrophages display anti-tumoural characteristics, whereas alternatively activated M2-macrophages are generally considered pro-tumoural. Both populations can co-exist in the tumour, although M2-like MHCII$^{low}$ TAMs are claimed to enrich in hypoxic and M1-like MHCII$^{hi}$ TAMs in normoxic areas[22,36]. Due to the complex environment of the gut, intestinal tissue-resident macrophages do

---

**Fig. 6** Tissue-resident macrophages in polyps expand by self-renewal and undergo metabolic reprogramming. **a** Proliferative activity of the LP and tumoural tissue-resident macrophage subpopulations. Intracellular expression levels of Ki-67 in adult F4/80$^{hi}$MHCII$^{hi}$ and F4/80$^{hi}$MHCII$^{low}$ cells, obtained from the colon LP of wild-type mice and colon tumours from dextran sodium sulphate-treated Apc$^{Min/+}$ mice. Representative dot plot and bar chart representing data obtain for colon LP ($n = 5$ mice) and tumours ($n = 10$ mice). Error bars represent the s.e.m. Statistical significance was determined by two-way ANOVA followed by Bonferroni test; **$P < 0.01$; ***$P < 0.001$; ns, not significant. **b, c** Myeloid subpopulations were sorted from pooled colon LP and tumours of 13-15 mice in three independent experiments in an RNA-seq study: Ly6C$^{hi}$MHCII$^-$ monocytes, Ly6G$^{hi}$F4/80$^-$CD11b$^+$ neutrophils, F4/80$^{hi}$MHCII$^{hi}$ and F4/80$^{hi}$MHCII$^{low}$ tissue-resident macrophage subpopulations. Gating strategy is shown in Supplementary Fig. 4. **b** Heat map and clustering of cell cycle-associated transcripts enriched in intratumoural F4/80$^{hi}$MHCII$^{hi}$ and F4/80$^{hi}$MHCII$^{low}$ subsets. The heat map was generated with log2 transformed RPKM values and with the row/gene median subtracted. **c** Heat map and clustering of glycolysis- and urea cycle-associated transcripts enriched in intratumoural F4/80$^{hi}$MHCII$^{hi}$ and F4/80$^{hi}$MHCII$^{low}$ subsets. The heat map was generated with log2 transformed FPKM values and with the row/gene median subtracted. **d** Quantitative PCR of Arg-1 expression levels in the LP and intratumoural F4/80$^{hi}$MHCII$^{hi}$ and F4/80$^{hi}$MHCII$^{low}$ subpopulations. The qPCR results shown are representative of sorted macrophage subpopulations obtained from pooled LP and tumours of 13-15 mice. Error bars represent the s.e.m. Statistical significance was determined by two-way ANOVA followed by Bonferroni test; ***$P < 0.001$ **e** Representative flow cytometry histograms (upper panels) and bar charts (lower panel) showing the intracellular mean of fluorescence intensity (MFI) of ARG1 expression in colon LP and intratumoural F4/80$^{hi}$MHCII$^{hi}$ and F4/80$^{hi}$MHCII$^{low}$ cells. Error bars represent the s.e.m. Statistical significance was determined by two-way ANOVA followed by Bonferroni test; **$P < 0.01$; ***$P < 0.001$; ns, not significant. **f** Differential expression of distinct metalloproteinases (Mmp2, Mmp9 and Mmp12) in LP and intratumoural F4/80$^{hi}$MHCII$^{hi}$ and F4/80$^{hi}$MHCII$^{low}$ subsets measured by qPCR. Error bars represent the s.e.m. Statistical significance was determined by two-way ANOVA followed by Bonferroni test; *$P < 0.05$; ***$P < 0.001$; ns, not significant. **g** Transcriptome analysis of distinct colon LP and tumoural myeloid cell populations. Hierarchical clustering of the LP and intratumoural monocytes (P1), neutrophils and tissue-resident macrophage (F4/80$^{hi}$MHCII$^{hi}$ and MHCII$^{low}$) subsets

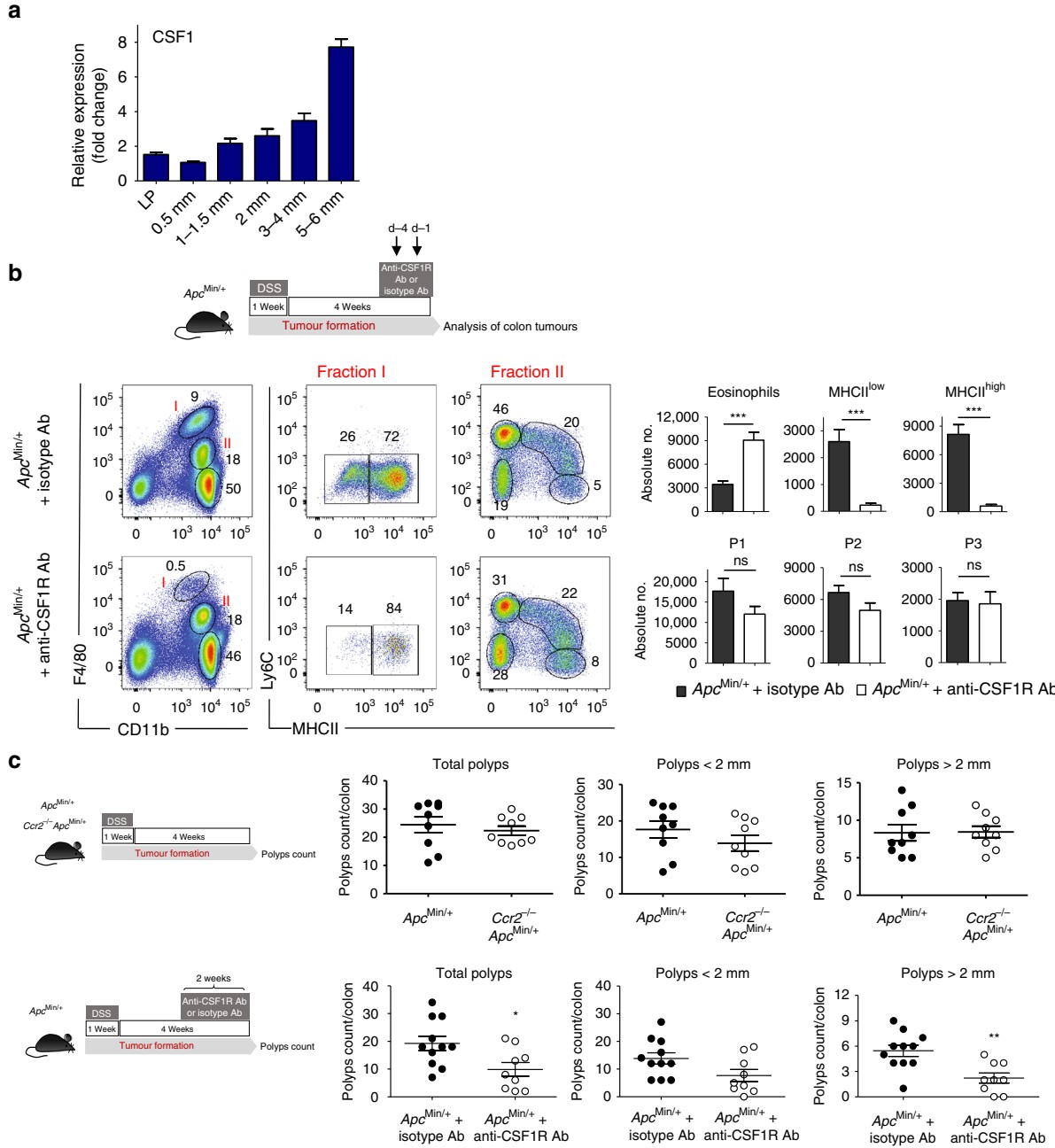

**Fig. 7** Pro-tumoural F4/80[hi] resident macrophages depend on CSF1. **a** Intra-tumoural CSF1 levels gradually increase with tumour progression. Colon LP and polyps of different sizes (ranging from 0.5 mm to 6 mm in diameter) were processed for quantitative PCR analysis of *Csf1* expression. Data were normalized to *β-actin* and are expressed as fold change in *Csf1* expression from 0.5 mm tumours. **b** Anti-CSF1 receptor (CSF1R) antibody depletes intratumoural F4/80[hi]MHCII[low] and F4/80[hi]MHCII[hi] macrophages. The 12-week-old dextran sodium sulphate (DSS)-treated *Apc*[Min/+] mice were injected intraperitoneally with rat IGg2a or anti-CSF1R blocking antibody (400 μg/mouse) at days −4 and −1 before collection of the colons, as shown in the schematic representation (upper panel). Polyps were processed and cells were analysed by flow cytometry for the presence of different myeloid cell populations defined in Fig. 1. Gating strategy is shown in Supplementary Fig. 4. The bar charts show the means±s.e.m. of absolute numbers of F4/80[hi]MHCII[hi] and F4/80[hi]MHCII[low] tissue-resident macrophage subpopulations, monocytes (P1), monocyte-derived macrophages (P2–P3) and eosinophils (control mice: *n* = 6 and anti-CSF1R Ab injected mice: *n* = 5). Statistical significance was determined using an unpaired Student's *t*-test. ***P < 0.0001; ns, not significant. **c** CSF1R blockade but not CCR2 deficiency reduces tumour burden in *Apc*[Min/+] mice. Polyp counts in the colon of *Apc*[Min/+] (*n* = 9) and *Ccr2*[−/−]*Apc*[Min/+] mice (*n* = 9) (upper panel) and anti-rat IgG2a injected *Apc*[Min/+] (*n* = 11) and anti-CSF1R Ab injected *Apc*[Min/+] mice (*n* = 9) (lower panel). Statistical significance was determined using an unpaired Student's *t*-test. *P < 0.05 and **P < 0.001

not follow the rigid M1–M2 classification[37] and "classic" M1 and M2 features are often shared by the same cells[20,38]. Similar mixed M1/M2 phenotype is observed in macrophages of colon tumours, since both F4/80[hi] MHCII[hi] and MHCII[low] populations express the classical M2 marker ARG1, but also many glycolytic genes typical for M1 macrophages, although the F4/80[hi]MHCII[low]-

resident macrophage subset shows always the highest transcript levels. As observed in other tumour models[22,36,39], F4/80[hi]MH-CII[low] cells, with their augmented metabolic switch together with their progressive predominance during tumour development, represent a pro-tumoural macrophage subpopulation also in intestinal adenomas.

Macrophages are not only crucial mediators of many pathologies (ranging from infectious to metabolic diseases and cancers) but also influence the outcome of anticancer therapeutics, such as chemotherapy, radiotherapy and even immunotherapy[40–42]. As such, these cells are becoming attractive candidates for therapeutic intervention. Manipulation of macrophage numbers[43] and/or reprogramming their phenotype[44] are techniques currently used to target macrophages in a therapeutic context. A better understanding of their ontogeny in different tumours, such as the data presented here for colon cancer, will facilitate the development of successful innovative macrophage-based therapeutic approaches.

## Methods

**Mice**. $Kit^{MerCreMer/R26}$ mice were generated in our laboratory as previously described[5]. $Ccr2^{-/-}$ (B6.129S4-$Ccr2^{tm1lfc}$/J)[45], C57BL/6J and $Apc^{Min/+}$ (C57BL/6J-ApcMin/J)[46] mice were obtained from The Jackson Laboratory (USA). $Apc^{Min/+}$ mice were backcrossed with $Kit^{MerCreMer/R26}$ mice or $Ccr2^{-/-}$ mice to obtain $Apc^{Min/+}Kit^{MerCreMer/R26}$ and $Ccr2^{-/-}Apc^{Min/+}$ mice, respectively. Mice were specific-pathogen-free (SPF) and maintained under a 12 h light/dark cycle with standard chow diet provided ad libitum. Both male and female mice (6–8 weeks old) were used for all experiments and were equally distributed within experimental and control groups.

All transgenic mice were bred and housed under SPF conditions in the Nanyang Technological University animal facility. This study was carried out in strict accordance with the recommendations of the NACLAR (National Advisory Committee for Laboratory Animal Research) guidelines under the Animal & Birds (Care and Use of Animals for Scientific Purposes) Rules of Singapore. The protocol was approved by the Institutional Animal Care and Use Committee of the Nanyang Technological University of Singapore.

**Spontaneous and chemical-induced intestinal tumourigenesis**. $Apc^{Min/+}$ mice spontaneously develop macroscopically detectable adenomas mainly in the small intestine and only few in the colon in aged mice[35]. Therefore, for all our experiments (with exception of the experiments shown in Fig. 2 and Supplementary Fig. 1) we opted for a chemically induced colitis model which selectively enhances the development of adenomas in the colon in a relatively short time of 5 weeks. Mice were terminated when showing symptoms of anaemia in combination with weight loss and/or other signs of physical discomfort.

In the "spontaneous" tumour model, mice were analysed at the age of 6–7 months, whereas in case of chemically induced tumourigenesis, 6–8-week-old female and male $Apc^{Min/+}$ mice (and correspondent control C57BL/6J mice) were treated with 1.5 or 1.25% (w/v) DSS (50,000 Da; MP Biomedicals, Santa Ana, CA, USA), respectively, in the drinking water for 1 week and sacrificed 4 weeks later at an age of 3–4 months.

**Colorectal polyp counts**. Polyps in each colon were macroscopically counted and categorized as >2 mm (large) and <2 mm (small). Colon tumours were measured ex vivo with the help of a sliding caliper.

**Tamoxifen-inducible fate mapping mouse model**. $Kit^{MerCreMer/R26}$ and $Apc^{Min/+}Kit^{MerCreMer/R26}$ mice were used for cell fate mapping to delineate the ontogeny of intratumoural macrophages. Then, 80 mg tamoxifen (Sigma, T5648; Sigma-Aldrich, St. Louis, MO, USA) was dissolved in 6 ml corn oil (C8267; Sigma) and a total of 4 mg tamoxifen per mouse was administered for five consecutive days by gavage for adult labelling or by intraperitoneal injection in pregnant mothers (embryonic stage 10.5) for embryonic labelling. After tamoxifen administration, c-kit+ BM-derived haematopoietic stem cells (HSCs) in $Apc^{Min/+}Kit^{MerCreMer/R26}$ mice were labelled with yellow fluorescent protein (YFP+) and all cells deriving from YFP+ BM-HSCs would maintain YFP expression. This effect enables to distinguish whether adult intratumoural macrophages are derived from adult BM-definitive haematopoiesis or from embryonic haematopoiesis. Adult and fetal labelling experimental protocols are shown in Fig. 5.

**Myeloid cell isolation**. LP myeloid cells were isolated as previously described[47]. Briefly, after $CO_2$ killing of the mice, the colon was opened and rinsed with phosphate-buffered saline (PBS) to remove the luminal contents. To remove the epithelium, the colon was incubated in 25 ml PBS with 1.3 mM EDTA under shaking conditions at 37 °C for 1 h. After incubation, the colon was washed in 2% Iscove's modified Dulbecco's medium (IMDM) to remove the EDTA solution, and then minced. The colon pieces were digested in 2% IMDM containing 1 mg/ml Collagenase D (Roche, Switzerland) under shaking conditions at 37 °C for 1.5 h. The digested tissue was then gently mashed through a 150 μm cell strainer. The leukocyte population was enriched using a 70/40% Percoll gradient (GE Healthcare Life Science, Chicago, IL, USA). Low-density cells at the interface were harvested and processed further for flow cytometry.

Tumours were manually cut, minced and digested in 2% IMDM containing 1 mg/ml Collagenase D and 20 U/ml DNase I (Life Technologies, Carlsbad, CA, USA) under shaking conditions at 37 °C for 1 h. Digested tissue was subsequently mashed through a cell strainer to obtain a cell suspension. The leukocyte population was enriched by 35% Percoll and further processed for flow cytometry and, if required, cell sorting using a FACSAria cell sorter (BD Biosciences, San Jose, CA, USA). Gating strategy used for sorting is shown in Supplementary Fig. 4.

**Transcriptomics analysis by RNA-sequencing**. Total RNA was extracted using the Arcturus™ PicoPure™ RNA Isolation Kit (Cat. No. KIT0214, Thermo Fisher Scientific, Waltham, MA, USA) according to the manufacturer's protocol. All mouse RNAs were analysed using an Agilent Bioanalyser (Agilent, Santa Clara, CA, USA) for quality assessment; the RNA Integrity Number range was 6.5–10, with a median of 9.2. Complementary DNA (cDNA) libraries were prepared from 2 ng total RNA starting material and 1 μl of a 1:50,000 dilution of ERCC RNA Spike in Controls (Cat. No. 4456740, Ambion® Thermo Fisher Scientific) using the SMARTSeq v2 protocol[48] with the following modifications: (1) addition of 20 μM TSO; and (2) use of 250 pg cDNA with 1/5 reaction of Illumina Nextera XT kit (Cat. No. FC-131−1024, Illumina, San Diego, CA, USA). The length distribution of the cDNA libraries was monitored using a DNA High Sensitivity Reagent Kit on the Perkin Elmer Labchip (Cat. No. CLS760672, Perkin Elmer, Waltham, MA, USA). All eight samples were subjected to an indexed paired-end sequencing run of 2 × 51 cycles on an Illumina HiSeq 2500 system (Illumina) under rapid run mode (17 samples/lane).

The paired-end reads were mapped to the Mouse GRCm38/mm10 reference genome using the STAR alignment tool[49]. Mapped reads were summarized to the gene level using featureCounts (V1.5.0-P1) software[50] and with GENCODE gene annotation[51]. Genes with an average number of reads per sample <10 in all cell subpopulations were filtered out from subsequent analyses. For differentially expressed gene (DEG) analysis, the limma/voom pipeline was used as recommended by the MicroArray Quality Control (MAQC) project[52] as one of the best performing RNA-seq data analysis pipelines. Comparisons between different cell populations were performed using limma and DEGs were selected with Benjamini–Hochberg adjusted $P$-values < 0.05. Hierarchical clustering and principal component analysis were performed with Log2 transformed value of RPKM (Reads Per Kilobase of transcript per Million mapped reads). All analyses were carried out in R version 3.1.2 (URL http://www.R-project.org/).

**Flow cytometry**. Single-cell suspensions were then stained and subsequently analysed by a BD Fortessa 5 laser flow cytometer (BD Bioscience). Data were analysed using a FlowJo software (TreeStar, Ashland, OR, USA). The following antibodies were used: APC/Cy7-labelled anti-CD11b (clone: M1/70, 1:600) and BV605-labelled anti-Ly6C (HK1.4, 1:600) were purchased from Biolegend (San Diego, CA, USA). PE-labelled anti-EMR1 (also known as F4/80) (clone: BM8, 1:600), PE/Cy7-labelled anti-CD45.2 (clone: 104, 1:600), APC-labelled anti-CD11c (clone: N418, 1:600) and eFluor®450-labelled anti-MHCII (clone:M5/114.15.2, 1:800) were obtained from eBioscience (San Diego, CA, USA). FITC-labelled polyclonal sheep anti-human/mouse ARG1 (1:20) and APC-labelled anti-CCR2 (Cat. No. FAB5538A) (1:20) were obtained from R&D System (Minneapolis, MN, USA), FITC-labelled anti-Mouse/Rat Ki-67 (clone SolA15, 1:40) from Miltenyi Biotech (Bergisch-Gladbach, Germany) and BUV395-labelled anti-Ly6G (clone: 1A8, 1:600) from BD (San Diego, CA, USA)

**Anti-CSF1R antibody treatment**. Twelve-week-old DSS-treated $Apc^{Min/+}$ mice were injected intraperitoneally with rat Ig2a isotype control (Biolegend) or anti-CSF1R blocking antibody (Clone AFS98, BioXCell, West Lebanon, NH, USA) (400 μg/mouse) at days −4 and −1 before collection of the colons.

**Quantitative PCR (qPCR) analysis**. cDNA was generated using SuperScript III Reverse Transcriptase (Cat. No. 18080093, Invitrogen, Carlsbad, CA, USA) according to the manufacturer's instructions. Quantitative real-time PCR was then performed using the FAST 2× qPCR Master mix (PrecisionFAST-SY, Primerdesign Ltd, Cambridge, UK). Reactions were run on a real-time qPCR system (Illumina, San Diego, CA, USA). Samples were normalized to β-actin, and data represent the mean of triplicate analyses and are displayed as a fold change from $F4/80^{hi}MHCII^{hi}$ LP macrophages unless otherwise stated. The primer sequences were as follows: $Arg1$; Fwd: gaatctgcgggcaacc, Rev: gaatcctggtacatctgggaac; $Mmp2$; Fwd: taacctg-gatgccgtcgt, Rev: ttcaggtaataagcacccttgaa; $Mmp9$; Fwd: catccagtatctgtatggtcgtg, Rev: gctgtggttcagttgtggtg; $Mmp12$; Fwd: gctgctcccatgaatgaca, Rev: aagcattgcacacggttgt; $Csf1$; Fwd: ggtggaactgccagtatagaaag, Rev: tcccatatgtctccttccataaa; $β$-actin Fwd: aaggccaaccgtgaaaagat, Rev: gtggtacgaccagaggcatac.

**Statistical analysis**. Statistical analysis was performed using GraphPad Prism 6 software (GraphPad Software, La Jolla, CA, USA). All values are expressed as the mean±s.e.m as indicated in the legend. Samples were analysed by unpaired Student's $t$-test (two-tailed) or Bonferroni two-way analysis of variance (ANOVA). A $P$-value of <0.05 was considered to be statistically significant.

**Data availability**. The accession number for the RNA-seq data reported in this paper is GEO: GSE90153. Original flow cytometry data are deposited in the NTU Open Access Data Repository DR-NTU (https://researchdata.ntu.edu.sg/dataset.xhtml?persistentId=doi:10.21979/N9/EQXCRF). All other data are available from the authors upon reasonable request.

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

## Acknowledgements

The authors would like to thank Monika Tetlak for providing excellent mouse management. The authors would also like to thank Insight Editing London for proofreading the manuscript prior to submission. This work was supported by MOE2014-T2-1-011 and MOE2016-T2-1-012 Ministry of Education Tier2 grants to C.R.

## Author contributions

Conceptualization: C.R.; methodology: I.S., J.S., S.F., J.L. and F.Z.; investigation: I.S., J.S. and Q.C.; formal analysis: I.S.; bioinformatic analysis: K.D. and M.P.; writing (original draft): C.R.; writing (review and editing): C.R. and K.K.; visualization: I.S.; funding acquisition: C.R.; supervision: J.S., K.K. and C.R.

## Additional information

**Competing interests:** The authors declare no competing financial interests.

