## [Peer Review File · Nature Communications]

Reviewers' comments:

Reviewer #1 (IBD, inflammation, gut cancer)(Remarks to the Author):

In the article entitled "The tumor microenvironment creates a niche for the self-renewal of resident macrophages in colon adenoma", Soncin and colleagues combine different mouse models of spontaneous tumorigenesis and lineage tracking to describe age-related and tumorigenesis-related kinetics of myeloid-derived cell populations in the colon. Their findings show that resident macrophages in the colon have a very low replacement rate by bone marrow-derived monocytes, and that they display increased self-renewal during tumorigenesis, in part due to an increase in CSF-1 expression in tumors. However, whereas this work puts considerable effort on the lineage characterization of two F4/80hi populations of resident macrophages, their role, importance, and/or involvement in tumorigenesis is not addressed. Furthermore, the way the experiments are explained, as well as the labeling of some graphs, are very confusing and misleading to the reader. Therefore, I consider this work needs to be extensively reviewed by the authors before publication.

Major comments

1 – While the characterization of the F4/80hi MHChi/MHClo populations is very interesting, especially the increase in MHClo macrophages during tumor growth, their role in their tumorigenesis model is completely unknown. Thus, it is difficult to ascertain if these populations may have important functions as tumor-associated macrophages (TAMs) or not. Indeed, TAMs can arise from Ly6Chi circulating monocytes, which correspond to the P1-3 populations defined by the authors in some of their experiments, and which in turn account for, at least, 50% of the total F4/80+ cells in the study. To deepen in the roles of these populations in tumorigenesis, I would suggest the authors to use their cDNA samples and their DNA libraries to characterize other pathways related to TAM functions supporting tumor growth, rather than focusing on intrinsic proliferation of macrophages.

2 – The explanations provided for most of the experiments are insufficient and misleading. For instance:

- What are the criteria the authors use to switch from DSS-accelerated tumorigenesis (figs 5-7) to spontaneous tumorigenesis (figs 2-3)? Because actually, there are important differences in the proportions of neutrophils and FrI macrophages. Do MHClo resident macrophages behave the same, in both models?

- Where is the information about CCR2-/- mice in the Methods section? We just know they were backcrossed to Apcmin/+ mice because it is stated in a figure legend. Besides, what are the animals defined as "WT" in fig. 3C? Does WT mean Apcmin/+? This is absolutely confusing and, no matter the answer, wrong. Finally, how are Apcmin/+ CCR2-/- mice phenotypically: do they have reduced tumor numbers? Reduced tumor size?

- In the experiments with CSF-1, does "WT" stand for "control" group, which should be Apcmin/+? Once again, this is extremely misleading for the reader. Why are animals of different ages receiving different treatments compared in this experiment? Finally, how do the authors interpret the sudden loss in resident tumor macrophages after 4 days of CSF-1 treatment: is it due to cell death? Disrupted proliferation? Are tumor sizes reduced by this treatment? This experiment needs further determinations to be performed.

3 – The number of determinations made in some of the experiments are insufficient. For instance, in fig. 2, the characterization of FrI macrophages at E19.5 was performed in a unique pool of 15 embryos. Since it is not specified if these embryos came from one or different mothers, it is unclear whether this finding may be representative or not. At least another pool of embryos should be included. Same problem occurs with results from tumors of 0.5 and 1 mm, which correspond respectively to 1 and 2 tumors.

4 – The use of statistics is not always correct. In figs. 5 and 6, one-way ANOVA cannot be used to compare the groups, as there are two variation factors: cell populations (MHChi vs MHClo) and original cell location (lamina propria vs tumor). Two-way ANOVA must be used, instead.

Minor comments

1 – The pie charts in fig. 1 and suppl. fig. 1 are misleading, as the authors only represent the CD11b+ cells that they characterized, which does not correspond to the total population of CD11b+ cells (as seen in bar graphs).

2 – The term “normal” is usually used mistakenly instead of “control”. For instance, in P5 L3 authors indicate they profiled subpopulations in normal, adult mouse colon, and refer fig. 1a. However, fig. 1a corresponds to mice that received DSS for 1 week. Even if the mice have 5 weeks to recover, in their flow cytometry it is possible to see that they have increased neutrophil and eosinophil proportions, when compared to WT mice in suppl. fig. 1a. Therefore, the authors should be careful and more exhaustive in their explanations about each model.

3 – In the figure legend for suppl. fig. 3, what do “a” and “b” sections stand for?

4 – In fig. 3C, are the absolute numbers for P1 correct?

5 – In P14 there is a repeated paragraph.

Reviewer #2 (Inflammation, macrophage)(Remarks to the Author):

In the current study, the authors demonstrate that intestinal tumor microenvironment supports self-renewal/proliferation of F4/80hiMHCIIlo myeloid cells normally found in neonates. In contrast, macrophage populations in healthy colon lamina propria (LP) tissue are continually replenished by circulating monocytes in a CCR2-dependent manner. The authors suggest that tumor-derived CSF-1 mediates this effect as anti-CSF-1R antibody reduces the numbers of F4/80hiMHCIIlo (and F4/80hiMHCIIhi) tumor-resident macrophages. Although the finding that the F4/80hiMHCIIlo cell population is maintained in tumors is interesting, the study is descriptive lacking any functional characterization.

Specific comments:

1. Given that surface MHCII expression is an indicator of macrophage functional polarization, the authors should conduct additional experiments to compare the similarity/difference between these two cell populations.

2. The physiological relevance of whether or how in situ macrophage proliferation is advantageous for tumor growth is not explored, which significantly reduces the impact of this study. The authors should address whether blocking F4/80hiMHCIIlo cells affect disease outcome.

Minor comment: p7. “Fig 3a and b” should be “Fig 3a and c”.

We indeed thank the reviewers for their constructive comments that make our data much stronger.

We have included amended Figure 3, 6 and 7 and a new Supplementary Fig. 3, which support our findings and solidify our results. All modifications in the text are marked in red.

Here our point-by-point reply.

Reviewer #1

Major comments

1 – While the characterization of the F4/80^{hi} MHC^{hi}/MHC^{lo} populations is very interesting, especially the increase in MHC^{lo} macrophages during tumor growth, their role in their tumorigenesis model is completely unknown. Thus, it is difficult to ascertain if these populations may have important functions as tumor-associated macrophages (TAMs) or not. Indeed, TAMs can arise from Ly6C^{hi} circulating monocytes, which correspond to the P1-3 populations defined by the authors in some of their experiments, and which in turn account for, at least, 50% of the total F4/80⁺ cells in the study. To deepen in the roles of these populations in tumorigenesis, I would suggest the authors to use their cDNA samples and their DNA libraries to characterize other pathways related to TAM functions supporting tumor growth, rather than focusing on intrinsic proliferation of macrophages.

A comprehensive analysis of our RNAseq data, Ingenuity Pathway Analysis (IPA[®]), qPCRs and flow cytometry analysis are now included in an amended Fig. 6 and a new Supplementary Fig. 3.

2 – The explanations provided for most of the experiments are insufficient and misleading. For instance:

- What are the criteria the authors use to switch from DSS-accelerated tumorigenesis (figs 5-7) to spontaneous tumorigenesis (figs 2-3)? Because actually, there are important differences in the proportions of neutrophils and FrI macrophages. Do MHC^{lo} resident macrophages behave the same, in both models?

Apc^{Min/+} mice spontaneously develop macroscopically detectable adenomas mainly in the small intestine and only few in the colon in aged mice hence it is commonly used to investigate tumor formation in small intestine but not in large intestine. Therefore, for all our experiments (with exception of the experiments shown in Fig. 2 and Supplementary Fig. 1) we opted for a chemically-induced colitis model which selectively enhances the development of adenomas in the colon in a relatively short time of 5 weeks. Considering that colon cancer is often linked with intestinal inflammation we are confident that this experimental animal model is suitable to investigate the myeloid landscape in the established tumors.

We included the “spontaneous” model (Fig. 2) mainly because it allowed us to monitor over a longer time period the progression of tumor. At 6 months of age adenomas can reach a size of 5-6 mm (average size in the DSS model 2-3 mm), which was important for our scoring of the MHCII^{hi/lo} ratio as a function of tumor size.

Since the spontaneous model is not the main approach used in our experiments, we did not analyse the myeloid subpopulations in a comprehensive RNAseq analysis.

We have now incorporated a detailed description of both “spontaneous” and DSS-accelerated tumor models and explained which approach was chosen for every experiment performed.

- Where is the information about CCR2^{-/-} mice in the Methods section? We just know they were backcrossed to Apc^{Min/+} mice because it is stated in a figure legend. Besides, what are the animals defined as “WT” in fig. 3C?

The description of the CCR2^{-/-} and Apc^{Min/+} CCR2^{-/-} mice is now included in the Materials and Methods section- Mice.

Does WT mean Apc^{min/+}? This is absolutely confusing and, no matter the answer, wrong.

Fig. 3 was amended (WT replaced by Apc^{Min/+}).

Finally, how are Apc^{Min/+} CCR2^{-/-} mice phenotypically: do they have reduced tumor numbers? Reduced tumor size?

Colonic polyp counts obtained from WT Apc^{Min/+} mice and Apc^{Min/+} CCR2^{-/-} mice were now included in a new Fig. 7c.

- In the experiments with CSF-1, does “WT” stand for “control” group, which should be Apc^{min/+}? Once again, this is extremely misleading for the reader.

We apologize for the confusion. WT was replaced by Apc^{Min/+} (Fig. 7).

Why are animals of different ages receiving different treatments compared in this experiment?

We apologize for the misunderstanding. The experiments were performed with 8-weeks old mice which were treated for one week with DSS and analysed 4 weeks later.

We have now corrected the paragraph in: 12-weeks old, dextran sodium sulphate (DSS)-treated APC^{Min/+} mice, injected with rat IgG2a (isotype control) or anti CSF1R Ab.

Finally, how do the authors interpret the sudden loss in resident tumor macrophages after 4 days of CSF-1 treatment: is it due to cell death? Disrupted proliferation?

It is well established that CSF-1 regulates the migration, proliferation, function and survival of macrophages hence the blockade of the CSF1/CSF1R axis can affect the numbers of these cells. Ries et al. (Cancer Cell 2014) shows that blocking the CSF1R reduced the viability of human macrophages differentiated with CSF-1 (IC₅₀ of 0.3 nM)

by inducing cell death. Like in our case with anti-CSF1R clone AFS98, the same authors demonstrate an anti-CSF1R-dependent depletion of tumor-associated macrophages both in mouse and humans.

Are tumor sizes reduced by this treatment? This experiment needs further determinations to be performed.

We have now included a new figure showing the number and sizes of the tumor polyps in mice treated for two weeks with anti-CSF1R antibody before harvesting and performing the polyps count (new Fig. 7 c)

3 – The number of determinations made in some of the experiments are insufficient. For instance, in fig. 2, the characterization of FrI macrophages at E19.5 was performed in a unique pool of 15 embryos. Since it is not specified if these embryos came from one or different mothers, it is unclear whether this finding may be representative or not. At least another pool of embryos should be included. Same problem occurs with results from tumors of 0.5 and 1 mm, which correspond respectively to 1 and 2 tumors.

The characterization of colon Fr1 macrophages was performed in a pool of 15 embryos obtained from three different pregnant mice (included now in the Legend of Fig. 2). The normalized labelling of 85-90% confirms our previously published data in Immunity 2016 (Sheng et al., where we demonstrated that colon Fr I cells become fully labelled during different stages of embryogenesis.

For better representation and statistical significance, we have included in our analysis some additional mice for 0.5 and 1 mm tumors (now included in the legend of Fig. 2).

4 – The use of statistics is not always correct. In figs. 5 and 6, one-way ANOVA cannot be used to compare the groups, as there are two variation factors: cell populations (MHChi vs MHClo) and original cell location (lamina propria vs tumor). Two-way ANOVA must be used, instead.

Two-Way ANOVA statistical analysis was performed for data shown in Fig. 5 and 6.

Minor comments

1 – The pie charts in fig. 1 and suppl. fig. 1 are misleading, as the authors only represent the CD11b⁺ cells that they characterized, which does not correspond to the total population of CD11b⁺ cells (as seen in bar graphs).

The PIE chart shows the proportions of the characterized myeloid cell populations and this was clarified in the corresponding Fig. Legend 1.

2 – The term “normal” is usually used mistakenly instead of “control”. For instance, in P5 L3 authors indicate they profiled subpopulations in normal, adult mouse colon, and refer fig. 1a. However, fig. 1a corresponds to mice that received DSS for 1 week. Even if the mice have 5 weeks to recover, in their flow cytometry it is possible to see that they have increased neutrophil and eosinophil proportions, when compared to WT mice in suppl. fig. 1a. Therefore, the authors should be careful and more exhaustive in their explanations about each

model.

3 – In the figure legend for suppl. fig. 3, what do “a” and “b” sections stand for?

The legends of Fig. 1 and Suppl. Fig. 1 were corrected. (a) stands for the dot plots of representative flow cytometry analyses (b) corresponding bar charts and (c) the PIE chart for proportions of myeloid cells in LP and colon adenomas.

4 – In fig. 3C, are the absolute numbers for P1 correct?

All absolute numbers were recalculated and correspondingly corrected.

5 – In P14 there is a repeated paragraph.

The duplicated paragraph was taken away.

Reviewer

#2

Specific

comments:

1. Given that surface MHCII expression is an indicator of macrophage functional polarization, the authors should conduct additional experiments to compare the similarity/difference between these two cell populations.

A comprehensive analysis of our RNAseq data, Ingenuity Pathway Analysis (IPA[®]), qPCRs and flow cytometry analysis are now included in an amended Fig. 6 and a new Supplementary Fig. 3.

2. The physiological relevance of whether or how in situ macrophage proliferation is advantageous for tumor growth is not explored, which significantly reduces the impact of this study. The authors should address whether blocking F4/80^{hi}MHCII^{lo} cells affect disease outcome.

We have now included additional results (Fig. 7c) which demonstrate the pro-tumoral properties of F4/80^{hi} macrophages. Polyp numbers and polyp size were analysed in anti-CSF1R Ab treated mice (lacking F4/80^{hi} macrophages) as well as, for comparison, in CCR2^{-/-}-APC^{Min/+} mice (lacking the CCR-2 dependent monocytes and monocyte-derived macrophages).

To our knowledge there is no specific animal model or antibody treatment to block or ablate specifically the F4/80^{hi} MHCII^{lo} fraction.

Accordingly, we have also changed the title in “The tumor microenvironment creates a niche for the self-renewal of tumor-promoting macrophages in colon adenoma”

Minor comment: p7. “Fig 3a and b” should be “Fig 3a and c”.

Corrected

REVIEWERS' COMMENTS:

Reviewer #1 (Remarks to the Author):

In the reviewed version of their article, Soncin and colleagues have included additional data supporting a role for F4/80hi MHCi and MHClo in promotion of tumor growth, such as metabolic profile changes associated to tumorigenesis, and the expression of TAM-associated genes (ARG1 and different MMPs). These data are further backed up by their final experiment showing that blockade of CSF-1, which basically affects these two populations of macrophages, reduces tumor burden, especially larger polyps. All my requests and comments have been met and properly addressed, and therefore I consider the article is ready for publication.

However, I still have a couple of minor comments:

- 1)The authors should state the percentage of DSS that the mice received, even the ones treated for 7 days. In the same vein, I consider that the term "healthy colon" used in the figure legend for Fig. 1 is not appropriate, as 4 days after a 7-day cycle of 2-3% DSS administration, the colon is usually in a recovery phase, with hyperproliferative crypts and increased inflammatory infiltration.
- 2)In the graph corresponding to the 0.5 mm tumors in Fig. 3, since the number of experiments was increased to 3, an error bar should be included.

Reviewer #2 (Remarks to the Author):

Authors have provided new data to support the relevance of the identified macrophage population in colon cancer progression. I have two comments on the new data.

1. The RNA-seq comparison between MHCII low and high is not particularly useful as both populations are CSF1-dependent. The anti-CSF1R ab experiment could not distinguish the contribution from the two populations. It will be more informative to compare P1 from FrII and MHCII low from tumors since the former is CCR2 dependent and does not seem to affect tumorigenesis.
2. Validation of RNA-seq data is very limited and should be expanded to include key metabolic pathways (as the authors suggest there is a metabolic shift). Raw RNA-seq data need to be provided/deposited.

Point-by-point reply to the reviewer's comments

Reviewer #1

In the reviewed version of their article, Soncin and colleagues have included additional data supporting a role for F4/80^{hi} MHC^{hi} and MHC^{lo} in promotion of tumor growth, such as metabolic profile changes associated to tumorigenesis, and the expression of TAM-associated genes (ARG1 and different MMPs). These data are further backed up by their final experiment showing that blockade of CSF-1, which basically affects these two populations of macrophages, reduces tumor burden, especially larger polyps. All my requests and comments have been met and properly addressed, and therefore I consider the article is ready for publication.

However, I still have a couple of minor comments:

1) The authors should state the percentage of DSS that the mice received, even the ones treated for 7 days. In the same vein, I consider that the term “healthy colon” used in the figure legend for Fig. 1 is not appropriate, as 4 days after a 7-day cycle of 2-3% DSS administration, the colon is usually in a recovery phase, with hyperproliferative crypts and increased inflammatory infiltration.

The DSS concentrations used in our experiments are clearly listed in the Materials and Methods section “Spontaneous and dextran sodium sulphate (DSS)-induced intestinal tumorigenesis in APC^{Min/+} mice” section (pages 16-17).

As suggested by the reviewer, the term “healthy” was taken away from the legend of Fig. 1.

2) In the graph corresponding to the 0.5 mm tumors in Fig. 3, since the number of experiments was increased to 3, an error bar should be included.

The correspondent figure (Fig. 2 not Fig. 3 as indicated by the reviewer) was corrected and the error bar included.

Reviewer#2

Authors have provided new data to support the relevance of the identified macrophage population in colon cancer progression. I have two comments on the new data.

1. The RNA-seq comparison between MHCII low and high is not particularly useful as both populations are CSF1-dependent. The anti-CSF1R ab experiment could not distinguish the contribution from the two populations. It will be more informative to compare P1 from FrII and MHCII low from tumors since the former is CCR2 dependent and does not seem to affect tumorigenesis.

The main aim of our RNAseq analysis shown in Fig. 6 was to demonstrate that resident colon F4/80^{hi} macrophages change their phenotype during tumor progression, becoming independent from a CCR2-mediated monocyte replacement, by undergoing a metabolic switch upregulating key genes involved in glycolysis, urea cycle and in the degradation of extracellular matrix. Furthermore, tumor F4/80^{hi} resident macrophages express higher levels of Arg-1 than their lamina propria counterparts, a prototype marker for pro-tumoral M2-type macrophages.

The reviewer #2 asked us to compare tumor F4/80^{hi}MHC^{low} and tumor monocytes (p1). We believe that this comparison does not bring further clarity to our manuscript.

Based on their ontogeny, phenotype and function, resident macrophages (both F4/80^{hi}MHC^{low} and F4/80^{hi}MHC^{high}) and monocytes (fraction p1) belong to different cell lineages, as also clearly documented in the PCA and dendrogram analysis (Fig. 6).

Not surprisingly, by performing the requested comparison, we have found more than 3000 differentially expressed genes (>2 fold change, GEO: GSE90153) between tumor F4/80^{hi}MHC^{low} and monocytes and the following Ingenuity Pathway Analysis (IPA) could, therefore, reveal almost 100 different pathways which were differentially regulated making the representation and interpretation of the data difficult.

Ultimately, and most importantly, the contribution of these two distinct cell populations in supporting tumor progression was demonstrated *in vivo* by analysing polyp numbers and their size in colons of mice lacking resident macrophages or CCR2-depend myeloid cells, respectively (Fig. 7).

2. Validation of RNA-seq data is very limited and should be expanded to include key metabolic pathways (as the authors suggest there is a metabolic shift). Raw RNA-seq data need to be provided/deposited.

Supplementary Fig. 3 includes an Ingenuity Pathway Analysis (IPA) between F4/80^{hi}MHC^{low} and F4/80^{hi}MHC^{high} populations and shows that multiple pathways involved in glycolysis, gluconeogenesis, urea cycle, HIF-1 and colorectal cancer metastasis signalling were significantly upregulated in F40/80^{hi}MHCII^{low} cells.

Raw RNAseq data were provided (page 21).

The accession number for the RNA-seq data reported in this paper is GEO: GSE90153